# A Benchmark for Interpretability Methods in Deep Neural Networks

**Sara Hooker,  Dumitru Erhan,  Pieter-Jan Kindermans,  Been Kim**
Google Brain
shooker,dumitru,pikinder,beenkim@google.com

## Abstract

We propose an empirical measure of the approximate accuracy of feature importance estimates in deep neural networks. Our results across several large-scale image classification datasets show that many popular interpretability methods produce estimates of feature importance that are not better than a random designation of feature importance. Only certain ensemble based approaches—VarGrad and SmoothGrad-Squared—outperform such a random assignment of importance. The manner of ensembling remains critical, we show that some approaches do no better then the underlying method but carry a far higher computational burden.

## 1  Introduction

In a machine learning setting, a question of great interest is estimating the influence of a given input feature to the prediction made by a model. Understanding what features are important helps improve our models, builds trust in the model prediction and isolates undesirable behavior. Unfortunately, it is challenging to evaluate whether an explanation of model behavior is reliable. First, there is no ground truth. If we knew what was important to the model, we would not need to estimate feature importance in the first place. Second, it is unclear which of the numerous proposed interpretability methods that estimate feature importance one should select [6, 5, 43, 30, 37, 33, 39, 36, 19, 22, 11, 9, 40, 32, 41, 27, 34, 2]. Many feature importance estimators have interesting theoretical properties e.g. preservation of relevance [5] or implementation invariance [37]. However even these methods need to be configured correctly [22, 37] and it has been shown that using the wrong configuration can easily render them ineffective [18]. For this reason, it is important that we build a framework to empirically validate the relative merits and reliability of these methods.

A commonly used strategy is to remove the supposedly informative features from the input and look at how the classifier degrades [29]. This method is cheap to evaluate but comes at a significant drawback. Samples where a subset of the features are removed come from a different distribution (as can be seen in Fig. 1). Therefore, this approach clearly violates one of the key assumptions in machine learning: the training and evaluation data come from the same distribution. Without re-training, it is unclear whether the degradation in model performance comes from the distribution shift or because the features that were removed are truly informative [9, 11].

For this reason we decided to verify how much information can be removed in a typical dataset before accuracy of a retrained model breaks down completely. In this experiment, we applied ResNet-50 [16], one of the most commonly used models, to ImageNet. It turns out that removing information is quite hard. With 90% of the inputs removed the network still achieves 63.53% accuracy compared to 76.68% on clean data. This implies that a strong performance degradation without re-training *might* be caused by a shift in distribution instead of removal of information.

Instead, in this work we evaluate interpretability methods by verifying how the accuracy of a retrained model degrades as features estimated to be important are removed. We term this approach **ROAR**, **R**em**O**ve **A**nd **R**etrain. For each feature importance estimator, ROAR replaces the fraction of the

pixels estimated to be most important with a fixed uninformative value. This modification (shown in Fig. 1) is repeated for each image in both the training and test set. To measure the change to model behavior after the removal of these input features, we separately train new models on the modified dataset such that train and test data comes from a similar distribution. More accurate estimators will identify as important input pixels whose subsequent removal causes the sharpest degradation in accuracy. We also compare each method performance to a *random* assignment of importance and a sobel edge filter [35]. Both of these control variants produce rankings that are independent of the properties of the model we aim to interpret. Given that these methods do not depend upon the model, the performance of these variants respresent a lower bound of accuracy that a interpretability method could be expected to achieve. In particular, a random baseline allows us to answer the question: *is the interpretability method more accurate than a random guess as to which features are important?* In Section 3 we will elaborate on the motivation and the limitations of ROAR.

We applied ROAR in a broad set of experiments across three large scale, open source image datasets: ImageNet [10], Food 101 [8] and Birdsnap [7]. In our experiments we show the following.

- Training performance is quite robust to removing input features. For example, after randomly replacing $90\%$ of all ImageNet input features, we can still train a model that achieves $63.53 \pm 0.13$ (average across 5 independent runs). This implies that a small subset of features are sufficient for the actual decision making. Our observation is consistent across datasets.

- The base methods we evaluate are no better or on par with a random estimate at finding the core set of informative features. However, we show that SmoothGrad-Squared (an unpublished variant of Classic SmoothGrad [34]) and Vargrad [2], methods which ensemble a set of estimates produced by basic methods, far outperform both the underlying method and a random guess. These results are consistent across datasets and methods.

- Not all ensemble estimators improve performance. Classic SmoothGrad [34] is worse than a single estimate despite being more computationally intensive.

## 2 Related Work

Interpretability research is diverse, and many different approaches are used to gain intuition about the function implemented by a neural network. For example, one can distill or constrain a model into a functional form that is considered more interpretable [4, 12, 38, 28]. Other methods explore the role of neurons or activations in hidden layers of the network [24, 26, 23, 42], while others use high level concepts to explain prediction results [17]. Finally there are also the input feature importance estimators that we evaluate in this work. These methods estimate the importance of an input feature for a specified output activation.

While there is no clear way to measure "correctness", comparing the relative merit of different estimators is often based upon human studies [30, 27, 21] which interrogate whether the ranking is meaningful to a human. Recently, there have been efforts to evaluate whether interpretability methods are *both* reliable and meaningful to human. For example, in [18] a unit test for interpretability methods is proposed which detects whether the explanation can be manipulated by factors that are not affecting the decision making process. Another approach considers a set of sanity checks that measure the change to an estimate as parameters in a model or dataset labels are randomized [2]. Closely related to this manuscript are the modification-based evaluation measures proposed originally by [29] with subsequent variations [19, 25]. In this line of work, one replaces the inputs estimated to be most important with a value considered meaningless to the task. These methods measure the subsequent degradation to the trained model at inference time. Recursive feature elimination methods [15] are a greedy search where the algorithm is trained on an iteratively altered subset of features. Recursive feature elimination does not scale to high dimensional datasets (one would have to retrain removing one pixel at a time) and unlike our work is a method to estimate feature importance (rather than evaluate different existing interpretability methods).

To the best of our knowledge, unlike prior modification based evaluation measures, our benchmark requires retraining the model from random initialization on the modified dataset rather than re-scoring the modified image at inference time. Without this step, we argue that one cannot decouple whether the model's degradation in performance is due to artifacts introduced by the value used to replace

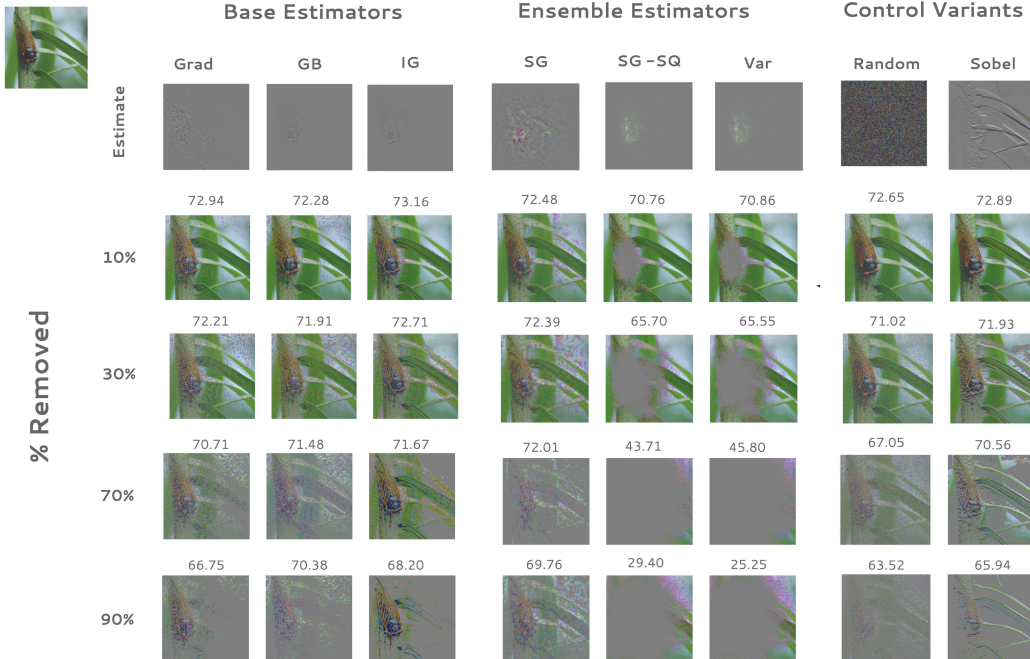

Figure 1: A single ImageNet image modified according to the ROAR framework. The fraction of pixels estimated to be most important by each interpretability method is replaced with the mean. Above each image, we include the average test-set accuracy for 5 ResNet-50 models independently trained on the modified dataset. **From left to right**: base estimators (gradient heatmap (GRAD), Integrated Gradients (IG), Guided Backprop (GB)), derivative approaches that ensemble a set of estimates (SmoothGrad Integrated Gradients (SG-SQ-IG), SmoothGrad-Squared Integrated Gradients (SG-SQ-IG), VarGrad Integrated Gradients (Var-IG)) and control variants (random modification (Random) and a sobel edge filter (Sobel)). This image is best visualized in digital format.

the pixels that are removed or due to the approximate accuracy of the estimator. Our work considers several large scale datasets, whereas all previous evaluations have involved a far smaller subset of data [3, 29].

## 3 ROAR: Remove And Retrain

To evaluate a feature importance estimate using ROAR, we sort the input dimensions according to the estimated importance. We compute an estimate $\mathbf{e}$ of feature importance for every input in the training and test set. We rank each $\mathbf{e}$ into an ordered set $\{e_i^o\}_{i=1}^N$. For the top $t$ fraction of this ordered set, we replace the corresponding pixels in the raw image with the per channel mean. We generate new train and test datasets at different degradation levels $t = [0., 10, \ldots, 100]$ (where $t$ is a percentage of all features modified). Afterwards the model is re-trained from random initialization on the new dataset and evaluated on the new test data.

Of course, because re-training can result in slightly different models, it is essential to repeat the training process multiple times to ensure that the variance in accuracy is low. To control for this, we repeat training 5 times for each interpretabiity method $\mathbf{e}$ and level of degradation $t$. We introduce the methodology and motivation for ROAR in the context of linear models and deep neural networks. However, we note that the properties of ROAR differ given an algorithm that explicitly uses feature selection (e.g. L1 regularization or any mechanism which limits the features available to the model at inference time). In this case one should of course mask the inputs that are known to be ignored by the model, *before* re-training. This will prevent them from being used after re-training, which could otherwise corrupt the ROAR metric. For the remainder of this paper, we focus on the performance of ROAR given deep neural networks and linear models which do not present this limitation.

**What would happen without re-training?** The re-training is the most computationally expensive aspect of ROAR. One should question whether it is actually needed. We argue that re-training is needed because machine learning models typically assume that the train and the test data comes from a similar distribution.

The replacement value $c$ can only be considered uninformative if the model is trained to learn it as such. Without retraining, it is unclear whether degradation in performance is due to the introduction of artifacts outside of the original training distribution or because we actually removed information. This is made explicit in our experiment in Section 4.3.1, we show that without retraining the degradation is far higher than the modest decrease in performance observed with re-training. This suggests retraining has better controlled for artefacts introduced by the modification.

**Are we evaluating the right aspects?** Re-training does have limitations. For one, while the architecture is the same, the model used during evaluation is not the same as the model on which the feature importance estimates were originally obtained. To understand why ROAR is still meaningful we have to think about what happens when the accuracy degrades, especially when we compare it to a random baseline. The possibilities are:

1. **We remove input dimensions and the accuracy drops.** In this case, it is very likely that the removed inputs were informative to the original model. ROAR thus gives a good indication that the importance estimate is of high quality.

2. **We remove inputs and the accuracy does not drop.** This can be explained as either:

   (a) It could be caused by removal of an input that was uninformative to the model. This includes the case where the input might have been informative but not in a way that is useful to the model, for example, when a linear model is used and the relation between the feature and the output is non-linear. Since in such a case the information was not used by the model and it does not show in ROAR we can assume ROAR behaves as intended.

   (b) There might be redundancy in the inputs. The same information could represented in another feature. This behavior can be detected with ROAR as we will show in our toy data experiment.

**Validating the behavior of ROAR on artificial data.** To demonstrate the difference between ROAR and an approach without re-training in a controlled environment we generate a 16 dimensional dataset with 4 informative features. Each datapoint $\mathbf{x}$ and its label $y$ was generated as follows:

$$\mathbf{x} = \frac{\mathbf{a}z}{10} + \mathbf{d}\eta + \frac{\epsilon}{10}, \qquad y = (z > 0).$$

All random variables were sampled from a standard normal distribution. The vectors $\mathbf{a}$ and $\mathbf{d}$ are 16 dimensional vectors that were sampled once to generate the dataset. In $\mathbf{a}$ only the first 4 values have nonzero values to ensure that there are exactly 4 informative features. The values $\eta, \epsilon$ were sampled independently for each example.

We use a least squares model as this problem can be solved linearly. We compare three rankings: the ground truth importance ranking, random ranking and the inverted ground truth ranking (the worst possible estimate of importance). In the left plot of Fig. 2 we can observe that without re-training the worst case estimator is shown to degrade performance relatively quickly. In contrast, ROAR shows no degradation until informative features begin to be removed at 75%. This correctly shows that this estimator has ranked feature importance poorly (ranked uninformative features as most important).

Finally, we consider ROAR performance given a set of variables that are completely redundant. We note that ROAR might not decrease until all of them are removed. To account for this we measure ROAR at different levels of degradation, with the expectation that across this interval we would be able to detect inflection points in performance that would indicate a set of redundant features. If this happens, we believe that it could be detected easily by the sharp decrease as shown in Fig. 2. Now that we have validated ROAR in a controlled setup, we can move on to our large scale experiments.

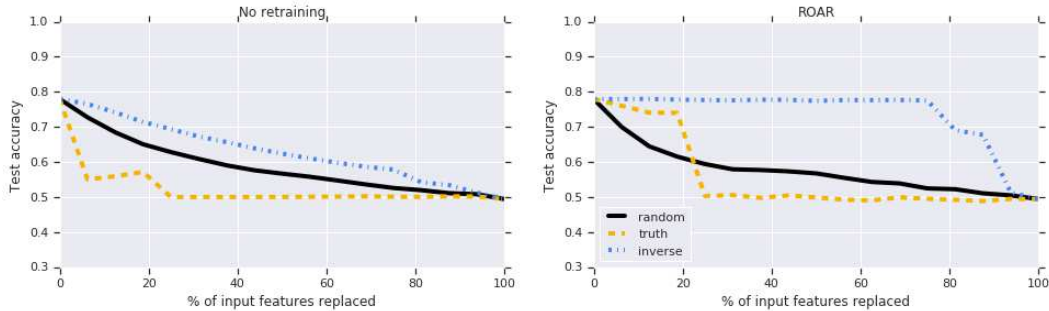

Figure 2: A comparison between not retraining and ROAR on artificial data. In the case where the model is not retrained, test-set accuracy quickly erodes despite the worst case ranking of redundant features as most important. This incorrectly evaluates a completely incorrect feature ranking as being informative. ROAR is far better at identifying this worst case estimator, showing no degradation until the features which are informative are removed at $75\%$. This plot also shows the limitation of ROAR, an accuracy decrease might not happen until a complete set of fully redundant features is removed. To account for this we measure ROAR at different levels of degradation, with the expectation that across this interval we would be able to control for performance given a set of redundant features.

## 4 Large scale experiments

### 4.1 Estimators under consideration

Our evaluation is constrained to a subset of estimators of feature importance. We selected these based on the availability of open source code, consistent guidelines on how to apply them and the ease of implementation given a ResNet-50 architecture [16]. Due to the breadth of the experimental setup it was not possible to include additional methods. However, we welcome the opportunity to consider additional estimators in the future, and in order to make it easy to apply ROAR to additional estimators we have open sourced our code `https://bit.ly/2ttLLZB`. Below, we briefly introduce each of the methods we evaluate.

**Base estimators** are estimators that compute a single estimate of importance (as opposed to ensemble methods). While we note that guided backprop and integrated gradients are examples of signal and attribution methods respectively, the performance of these estimators should not be considered representative of other methods, which should be evaluated separately.

- **Gradients or Sensitivity heatmaps [33, 6] (GRAD)** are the gradient of the output activation of interest $A_n^l$ with respect to $x_i$:

$$\mathbf{e} = \frac{\partial A_n^l}{\partial x_i}$$

- **Guided Backprop [36] (GB)** is an example of a signal method that aim to visualize the input patterns that cause the neuron activation $A_n^l$ in higher layers [36, 39, 19]. GB computes this by using a modified backpropagation step that stops the flow of gradients when less than zero at a ReLu gate.

- **Integrated Gradients [37] (IG)** is an example of an attribution method which assign importance to input features by decomposing the output activation $A_n^l$ into contributions from the individual input features [5, 37, 22, 31, 19]. Integrated gradients interpolate a set of estimates for values between a non-informative reference point $\mathbf{x}^0$ to the actual input $\mathbf{x}$. This integral can be approximated by summing a set of $k$ points at small intervals between $\mathbf{x}^0$ and $\mathbf{x}$:

$$\mathbf{e} = (\mathbf{x}_i - \mathbf{x}_i^0) \times \sum_{i=1}^{k} \frac{\partial f_w(\mathbf{x}^0 + \frac{i}{k}(\mathbf{x} - \mathbf{x}^0))}{\partial \mathbf{x_i}} \times \frac{1}{k}$$

The final estimate $\mathbf{e}$ will depend upon both the choice of $k$ and the reference point $\mathbf{x}^0$. As suggested by [37], we use a black image as the reference point and set $k$ to be 25.

**Ensembling methods** In addition to the base approaches we also evaluate three ensembling methods for feature importance. For all the ensemble approaches that we describe below (SG, SG-SQ, Var), we average over a set of 15 estimates as suggested by in the original SmoothGrad publication [34].

- **Classic SmoothGrad (SG) [34]** SG averages a set $J$ noisy estimates of feature importance (constructed by injecting a single input with Gaussian noise $\eta$ independently $J$ times):

$$\mathbf{e} = \sum_{i=1}^{J} (g_i(\mathbf{x} + \eta, A_n^l))$$

- **SmoothGrad$^2$(SG-SQ)** is an unpublished variant of classic SmoothGrad SG which squares each estimate $\mathbf{e}$ before averaging the estimates:

$$\mathbf{e} = \sum_{i=1}^{J} (g_i(\mathbf{x} + \eta, A_n^l)^2)$$

  Although SG-SQ is not described in the original publication, it is the default open-source implementation of the open source code for SG: `https://bit.ly/2Hpx5ob`.

- **VarGrad (Var) [2]** employs the same methodology as classic SmoothGrad (SG) to construct a set of t $J$ noisy estimates. However, VarGrad aggregates the estimates by computing the variance of the noisy set rather than the mean.

$$\mathbf{e} = \text{Var}(g_i(\mathbf{x} + \eta, A_n^l))$$

**Control Variants** As a control, we compare each estimator to two rankings (a random assignment of importance and a sobel edge filter) that do not depend at all on the model parameters. These controls represent a lower bound in performance that we would expect all interpretability methods to outperform.

- **Random** A random estimator $g^R$ assigns a random binary importance probability $\mathbf{e} \mapsto 0, 1$. This amounts to a binary vector $\mathbf{e} \sim Bernoulli(1-t)$ where $(1-t)$ is the probability of $e_i = 1$. The formulation of $g^R$ does not depend on either the model parameters or the input image (beyond the number of pixels in the image).

- **Sobel Edge Filter** convolves a hard-coded, separable, integer filter over an image to produce a mask of derivatives that emphasizes the edges in an image. A sobel mask treated as a ranking $\mathbf{e}$ will assign a high score to areas of the image with a high gradient (likely edges).

### 4.2 Experimental setup

We use a ResNet-50 model for both generating the feature importance estimates and subsequent training on the modified inputs. ResNet-50 was chosen because of the public code implementations (in both PyTorch [14] and Tensorflow [1]) and because it can be trained to give near to state of art performance in a reasonable amount of time [13].

For all train and validation images in the dataset we first apply test time pre-processing as used by Goyal et al. [13]. We evaluate ROAR on three open source image datasets: ImageNet, Birdsnap and Food 101. For each dataset and estimator, we generate new train and test sets that each correspond to a different fraction of feature modification $t = [0, 10, 30, 50, 70, 90]$. We evaluate 18 estimators in total (this includes the base estimators, a set of ensemble approaches wrapped around each base and finally a set of squared estimates). In total, we generate 540 large-scale modified image datasets in order to consider all experiment variants (180 new test/train for each original dataset).

We independently train 5 ResNet-50 models from random initialization on each of these modified dataset and report test accuracy as the average of these 5 runs. In the base implementation, the ResNet-50 trained on an unmodified ImageNet dataset achieves a mean accuracy of $76.68\%$. This is comparable to the performance reported by [13]. On Birdsnap and Food 101, our unmodified datasets achieve $66.65\%$ and $84.54\%$ respectively (average of 10 independent runs). This baseline performance is comparable to that reported by Kornblith et al. [20].

### 4.3 Experimental results

#### 4.3.1 Evaluating the random ranking

Comparing estimators to the random ranking allows us to answer the question: *is the estimate of importance more accurate than a random guess?* It is firstly worthwhile noting that model performance is remarkably robust to random modification. After replacing a large portion of all inputs with a constant value, the model not only trains but still retains most of the original predictive power. For example, on ImageNet, when only $10\%$ of all features are retained, the trained model still attains $63.53\%$ accuracy (relative to unmodified baseline of $76.68\%$). The ability of the model to extract a meaningful representation from a small random fraction of inputs suggests a case where many inputs are likely redundant. The nature of our input–an image where correlations between pixels are expected – provides one possible readons for redundancy.

The results for our random baseline provides additional support for the need to re-train. We can compare random ranking on ROAR vs. a traditional deletion metric [29], i.e. the setting where we do not retain. These results are given in Fig. 3. Without retraining, a random modification of $90\%$ degrades accuracy to $0.5\%$ for the model that was not retrained. Keep in mind that on clean data we achieve $76.68\%$ accuracy. This large discrepancy illustrates that without retraining the model, it is not possible to decouple the performance of the ranking from the degradation caused by the modification itself.

#### 4.3.2 Evaluating Base Estimators

Now that we have established the baselines, we can start evaluating the base estimators: GB, IG, GRAD. Surprisingly, the left inset of Fig. 4 shows that these estimators consistently perform worse than the random assignment of feature importance across all datasets and for all thresholds $t = [0.1, 0.3, 0.5, 0.7, 0.9]$. Furthermore, our estimators fall further behind the accuracy of random guess as a larger fraction $t$ of inputs is modified. The gap is widest when $t = 0.9$.

Our base estimators also do not compare favorably to the performance of a sobel edge filter SOBEL. Both the sobel filter and the random ranking have formulations that are entirely independent of the model parameters. All the base estimators that we consider have formulations that depend upon the trained model weights, and thus we would expect them to have a clear advantage in outperforming the control variants. However, across all datasets and thresholds $t$, the base estimators GB, IG, GRAD perform on par or worse than SOBEL.

Base estimators perform within a very narrow range. Despite the very different formulations of base estimators that we consider, the difference between the performance of the base estimators is in a strikingly narrow range. For example, as can be seen in the left column of Fig. 4, for Birdsnap, the difference in accuracy between the best and worst base estimator at $t = 90\%$ is only $4.22\%$. This range remains narrow for both Food101 and ImageNet, with a gap of $5.17\%$ and $3.62$ respectively. Our base estimator results are remarkably consistent results across datasets, methods and for all fractions of $t$ considered. The variance is very low across independent runs for all datasets and estimators. The maximum variance observed for ImageNet was a variance of $1.32\%$ using SG-SQ-GRAD at $70\%$ of inputs removed. On Birdsnap the highest variance was $0.12\%$ using VAR-GRAD at $90\%$ removed. For food101 it was $1.52\%$ using SG-SQ-GRAD at $70\%$ removed.

Finally, we compare performance of the base estimators using ROAR re-training vs. a traditional deletion metric [29], again the setting where we do not retain. In Fig. 3 we see a behavior for the base estimators on all datasets that is similar to the behavior of the inverse (worst possible) ranking on the toy data in Fig. 2. The base estimators appear to be working when we do not retrain, but they are clearly not better than the random baseline when evaluated using ROAR. This provides additional support for the need to re-train.

#### 4.3.3 Evaluating Ensemble Approaches

Since the base estimators do not appear to perform well, we move on to ensemble estimators. Ensemble approaches inevitably carry a higher computational approach, as the methodology requires the aggregation of a set of individual estimates. However, these methods are often preferred by humans because they appear to produce "less noisy" explanations. However, there is limited theoretical understanding of what these methods are actually doing or how this is related to the accuracy of the

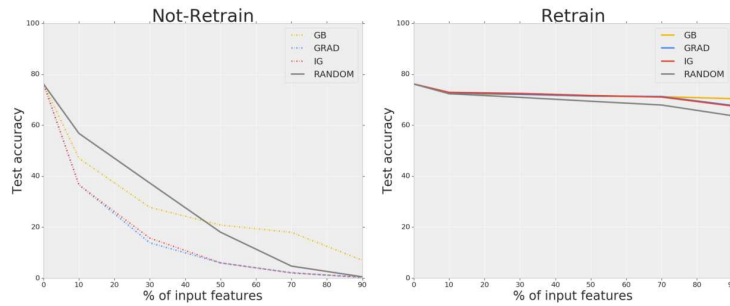

Figure 3: On the left we evaluate three base estimators and the random baseline without retraining. All of the methods appear to reduce accuracy at quite a high rate. On the right, we see, using ROAR, that after re-training most of the information is actually still present. It is also striking that in this case the base estimators perform worse than the random baseline.

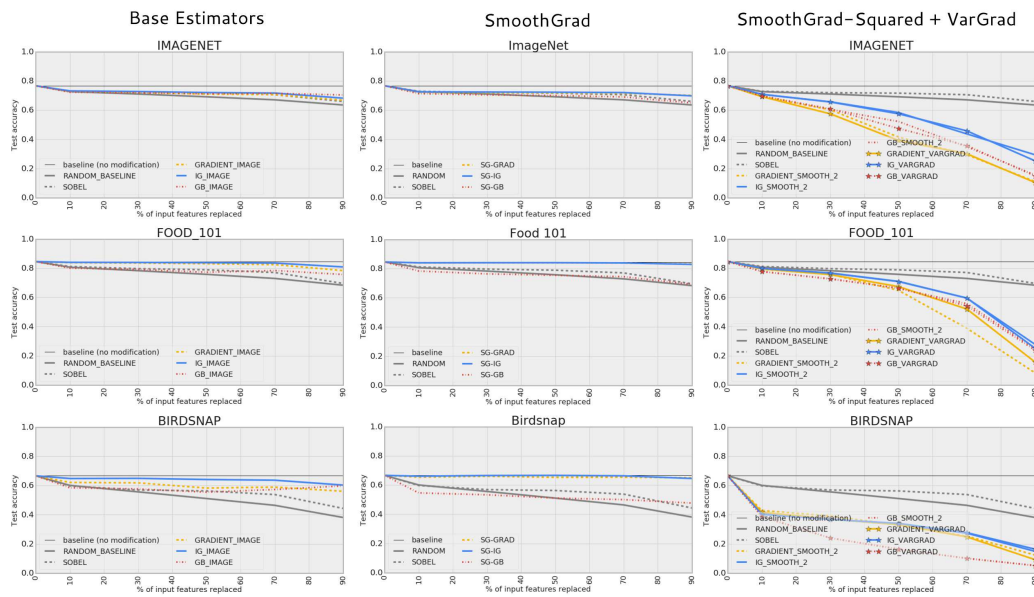

Figure 4: **Left:** Grad (GRAD), Integrated Gradients (IG) and Guided Backprop (GB) perform worse than a random assignment of feature importance. **Middle:** SmoothGrad (SG) is less accurate than a random assignment of importance and often worse than a single estimate (in the case of raw gradients SG-Grad and Integrated Gradients SG-IG). **Right:** SmoothGrad Squared (SG-SQ) and VarGrad (VAR) produce a dramatic improvement in approximate accuracy and far outperform the other methods in all datasets considered, regardless of the underlying estimator.

explanation. We evaluate ensemble estimators and produce results that are remarkably consistent results across datasets, methods and for all fractions of $t$ considered.

**Classic Smoothgrad** is less accurate or on par with a single estimate. In the middle column in Fig. 4 we evaluate Classic SmoothGrad (SG). It average 15 estimates computed according to an underlying base method (GRAD, IG or GB). However, despite averaging SG degrades test-set accuracy still less than a random guess. In addition, for GRAD and IG SmoothGrad performs worse than a single estimate.

**SmoothGrad-Squared and VarGrad** produce large gains in accuracy. In the right inset of Fig. 4, we show that both VarGrad (VAR) and SmoothGrad-Squared (SG-SQ) far outperform the two control variants. In addition, for all the interpretability methods we consider, VAR or SG-SQ far outperform the approximate accuracy of a single estimate. However, while VAR and SG-SQ benefit the accuracy of all base estimators, the overall ranking of estimator performance differs by dataset. For ImageNet and Food101, the best performing estimators are VAR or SG-SQ when wrapped around GRAD. However, for the Birdsnap dataset, the most approximately accurate estimates are these ensemble approaches wrapped around GB. This suggests that while the VAR and SG-SQ consistently improve performance, the choice of the best underlying estimator may vary by task.

Now, why do both of these methods work so well? First, these methods are highly similar. If the average (squared) gradient over the noisy samples is zero then VAR and SG-SQ reduce to the same method. For many images it appears that the mean gradient is much smaller than the mean squared gradient. This implies that the final output should be similar. Qualitatively this seems to be the case as well. In Fig. 1 we observe that both methods appear to remove whole objects. The other methods removed inputs that are less concentrated but spread more widely over the image. It is important to note that these methods were not forced to behave as such. It is emergent behavior. Understanding why this happens and why this is beneficial should be the focus of future work.

**Squaring estimates** The final question we consider is why SmoothGrad-Squared SG-SQ dramatically improves upon the performance of SmoothGrad SG despite little difference in formulation. The only difference between the two estimates is that SG-SQ squares the estimates before averaging. We consider the effect of only squaring estimates (no ensembling). We find that while squaring improves the accuracy of all estimators, the transformation does not adequately explain the large gains that we observe when applying VAR or SG-SQ. When base estimators are squared they slightly outperform the random baseline (all results included in the supplementary materials).

## 5   Conclusion and Future Work

In this work, we propose ROAR to evaluate the quality of input feature importance estimators. Surprisingly, we find that the commonly used base estimators, Gradients, Integrated Gradients and Guided BackProp are worse or on par with a random assignment of importance. Furthermore, certain ensemble approaches such as SmoothGrad are far more computationally intensive but do not improve upon a single estimate (and in some cases are worse). However, we do find that VarGrad and SmoothGrad-Squared strongly improve the quality of these methods and far outperform a random guess. While the low effectiveness of many methods could be seen as a negative result, we view the remarkable effectiveness of SmoothGrad-Squared and VarGrad as important progress within the community. Our findings are particularly pertinent for sensitive domains where the accuracy of a explanation of model behavior is paramount. While we venture some initial consideration of why certain ensemble methods far outperform other estimator, the divergence in performance between the ensemble estimators is an important direction of future research.

**Acknowledgments**

We thank Gabriel Bender, Kevin Swersky, Andrew Ross, Douglas Eck, Jonas Kemp, Melissa Fabros, Julius Adebayo, Simon Kornblith, Prajit Ramachandran, Niru Maheswaranathan, Gamaleldin Elsayed, Hugo Larochelle, Varun Vasudevan for their thoughtful feedback on earlier iterations of this work. In addition, thanks to Sally Jesmonth, Dan Nanas and Alexander Popper for institutional support and encouragement.

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
