[Supplementary Material]

# 1 Supplementary Charts and Experiments

We include supplementary experiments and additional details about our training procedure, the estimators we evaluate, the image modification process and test-set accuracy below. In addition, as can be seen in Fig. 1, we also consider the scenario where pixels are kept according to importance rather than removed.

## 1.1 Training Procedure

We carefully tuned the hyperparamters of each dataset ImageNet, Birdsnap and Food 101 separately. We find that the Birdsnap and Food 101 converge within the same amount of training steps and a larger learning rate than ImageNet. These are detailed in Table. 1.2. These hyper parameters, along with the mean accuracy reported on the unmodified dataset, are used consistently across all estimators. ImageNet dataset achieves a mean accuracy of $76.68\%$. This is comparable to the performance reported by [? ]. On Birdsnap and Food 101, our unmodified datasets achieve $66.65\%$ and $84.54\%$ respectively. The baseline test-set accuracy for Food101 or Birdsnap is comparable to that reported by [? ]. In Table. 2, we include the test-set performance for each experiment variant that we consider. The test-set accuracy reported is the average of $5$ independent runs.

Figure 1: Evaluation of all estimators according to Keep and Retrain KAR vs. ROAR. **Left inset:** For KAR, **K**eep **A**nd **R**etrain, we keep a fraction of features estimated to be most important and replace the remaining features with a constant mean value. The most accurate estimator is the one that preserves model performance the most for a given fraction of inputs removed (the highest test-set accuracy).**Right inset:** For **ROAR**, **R**em**0**ve **A**nd **Retrain** we remove features by replacing a fraction of the inputs estimated to be most important according to each estimator with a constant mean value. The most accurate estimator is the one that degrades model performance the most for a given fraction of inputs removed. Inputs modified according to KAR result in a very narrow range of model accuracy. ROAR is a more discriminative benchmark, which suggests that retaining performance when the most important pixels are removed (rather than retained) is a harder task.

Figure 2: Certain transformations of the estimate can substantially improve accuracy of all estimators. Squaring alone provides small gains to the accuracy of all estimators, and is slightly better than a random guess. **Left inset:** The three base estimators that we consider (Gradients (GRAD), Integrated Gradients (IG) and Guided Backprop (GB)) perform worse than a random assignment of feature importance. At all fractions considered, a random assignment of importance degrades performance more than removing the pixels estimated to be most important by base methods. **Right inset:** Average test-set accuracy across 5 independent iterations for estimates that are squared before ranking and subsequent removal. When squared, base estimators perform slightly better than a random guess. However, this does not compare to the gains in accuracy of averaging a set of noisy estimates that are squared (SmoothGrad-Squared)

| Dataset | Top1Accuracy | Train Size | Test Size | Learning Rate | Training Steps |
|---------|-------------|-----------|-----------|---------------|----------------|
| Birdsnap | 66.65 | 47,386 | 2,443 | 1.0 | 20,000 |
| Food_101 | 84.54 | 75,750 | 25,250 | 0.7 | 20,000 |
| ImageNet | 76.68 | 1,281,167 | 50,000 | 0.1 | 32,000 |

Table 1: The training procedure was carefully finetuned for each dataset. These hyperparameters are consistently used across all experiment variants. The baseline accuracy of each unmodified data set is reported as the average of 10 independent runs.

## 1.2 Generation of New Dataset

We evaluate ROAR on three open source image datasets: ImageNet, Birdsnap and Food 101. For each dataset and estimator, we generate 10 new train and test sets that each correspond to a different fraction of feature modification $t = [0.1, 0.3, 0.5, 0.7, 0.9]$ and whether the most important pixels are removed or kept. This requires first generating a ranking of input importance for each input image according to each estimator. All of the estimators that we consider evaluate feature importance post-training. Thus, we generate the rankings according to each intepretability method using a stored checkpoint for each dataset.

Figure 3: A single example from each dataset generated from modifying Food 101 according to both ROAR and KAR. We show the modification for base estimators (Integrated Gradients (IG), Guided Backprop (GB), Gradient Heatmap (GRAD) and derivative ensemble approaches - SmoothGrad, (SG-GRAD, SG-IG, SG-GB), SmoothGrad-Squared (SG-SQ-GRAD, SG-SQ-IG, SG-SQ-GB) and VarGrad (VAR-GRAD, VAR-IG, VAR-GB. In addition, we consider two control variants a random baseline, a sobel edge filter.

We use the ranking produced by the interpretability method to modify each image in the dataset (both train and test). We rank each estimate, $e$ into an ordered set $\{e_i^o\}_{i=1}^N$. For the top $t$ fraction of this ordered set, we replace the corresponding pixels in the raw image with a per channel mean. Fig. 3 and Fig. 4 show an example of the type of modification applied to each image in the dataset for Birdsnap and Food 101 respectively. In the paper itself, we show an example of a single image from each ImageNet modification.

We evaluate 18 estimators in total (this includes the base estimators, a set of ensemble approaches wrapped around each base and finally a set of squared estimates). In total, we generate 540 large-scale modified image datasets in order to consider all experiment variants (180 new test/train for each original dataset).

## 1.3 Evaluating Keeping Rather Than Removing Information

In addition to ROAR, as can be seen in Fig. 1, we evaluate the opposite approach of **KAR**, **K**eep **A**nd **R**etrain. While ROAR removes features by replacing a fraction of inputs estimated to be most

Figure 4: A single example from each dataset generated from modifying Imagenet according to the ROAR and KAR. We show the modification for base estimators (Integrated Gradients (IG), Guided Backprop (GB), Gradient Heatmap (GRAD) and derivative ensemble approaches - SmoothGrad, (SG-GRAD, SG-IG, SG-GB), SmoothGrad-Squared (SG-SQ-GRAD, SG-SQ-IG, SG-SQ-GB) and VarGrad (VAR-GRAD, VAR-IG, VAR-GB. In addition, we consider two control variants a random baseline, a sobel edge filter.

important, KAR preserves the inputs considered to be most important. Since we keep the important information rather than remove it, *minimizing* degradation to test-set accuracy is desirable.

In the right inset chart of Fig. 1 we plot KAR on the same curve as ROAR to enable a more intuitive comparison between the benchmarks. The comparison suggests that KAR appears to be a poor discriminator between estimators. The x-axis indicates the fraction of features that are preserved/removed for KAR/ROAR respectively.

We find that KAR is a far weaker discriminator of performance; all base estimators and the ensemble variants perform in a similar range to each other. These findings suggest that the task of identifying features to preserve is an easier benchmark to fulfill than accurately identifying a fraction of input that will cause the maximum damage to the model performance.

## 1.4 Squaring Alone Slightly Improves the Performance of All Base Variants

The surprising performance of SmoothGrad-Squared (SG-SQ) deserves further investigation; why is averaging a set of squared noisy estimates so effective at improving the accuracy of the ranking? To disentangle whether both squaring and then averaging are required, we explore whether we achieve similar performance gains by **only** squaring the estimate.

Squaring of a single estimate, with no ensembling, benefits the accuracy of all estimators that we considered. In the right inset chart of Fig. 2, we can see that squared estimates perform better than the raw estimate. When squared, an estimate gains slightly more accuracy than a random ranking of input features. In particular, squaring benefits GB; at $t = .9$ performance of SQ-GB relative to GB improves by $8.43\% \pm 0.97$.

Squaring is an equivalent transformation to taking the absolute value of the estimate before ranking all inputs. After squaring, negative estimates become positive, and the ranking then only depends upon the magnitude and not the direction of the estimate. The benefits gained by squaring furthers our understanding of how the direction of GB, IG and GRAD values should be treated. For all these estimators, estimates are very much a reflection of the weights of the network. The magnitude may be far more telling of feature importance than direction; a negative signal may be just as important as positive contributions towards a model's prediction. While squaring improves the accuracy of all estimators, the transformation does not explain the large gains in accuracy that we observe when we average a set of noisy squared estimates.

## 1.5 Limitations on the use of ROAR

In this work we propose ROAR as a method for estimating feature importance in deep neural networks. However, we do note that ROAR is not suitable for certain algorithms such as decision stump Y=(A or D) where there is also feature redundancy. For these algorithms, in order to use ROAR correctly feature importance must be recomputed after each re-training step. This is because a decision stump ignores a subset of input features at inference time which means it is possible for a random estimator to appear to perform better than the best possible estimator. For the class of models evaluated in the paper (linear models, multi-layer perceptrons and deep neural networks) as well as any model that allows all features to contribute to a prediction at test time ROAR remains valid. To make ROAR valid for decisions stumps, one can re-compute the feature importance after each re-training step. The scale of our experiments preclude this, and our experiments show that it is not necessary for deep neural networks (DNNs).

| | Threshold | Keep | | | | | Remove | | | | |
|---|---|---|---|---|---|---|---|---|---|---|---|
| | | 10.0 | 30.0 | 50.0 | 70.0 | 90.0 | 10.0 | 30.0 | 50.0 | 70.0 | 90.0 |
| **Birdsnap** | Random | 37.24 | 46.41 | 51.29 | 55.38 | 59.92 | 60.11 | 55.65 | 51.10 | 46.45 | 38.12 |
| | Sobel | 44.81 | 52.11 | 55.36 | 55.69 | 59.08 | 59.73 | 56.94 | 56.30 | 53.82 | 44.33 |
| | GRAD | 57.51 | 61.10 | 60.79 | 61.96 | 62.49 | 62.12 | 61.82 | 58.29 | 58.91 | 56.08 |
| | IG | 62.64 | 65.02 | 65.42 | 65.46 | 65.50 | 64.79 | 64.91 | 64.12 | 63.64 | 60.30 |
| | GP | 62.59 | 62.35 | 60.76 | 61.78 | 62.44 | 58.47 | 57.64 | 55.47 | 57.28 | 59.76 |
| | SG-GRAD | 64.64 | 65.87 | 65.32 | 65.49 | 65.78 | 65.44 | 66.08 | 65.33 | 65.44 | 65.02 |
| | SG-IG | 65.36 | 66.45 | 66.38 | 66.37 | 66.35 | 66.11 | 66.56 | 66.65 | 66.37 | 64.54 |
| | SG-GB | 52.86 | 56.44 | 58.32 | 59.20 | 60.35 | 54.67 | 53.37 | 51.13 | 50.07 | 47.71 |
| | SG-SQ-GRAD | 55.32 | 60.79 | 62.13 | 63.63 | 64.99 | 42.88 | 39.14 | 32.98 | 25.34 | 12.40 |
| | SG-SQ-IG | 55.89 | 61.02 | 62.68 | 63.63 | 64.43 | 40.85 | 36.94 | 33.37 | 27.38 | 14.93 |
| | SG-SQ-GB | 49.32 | 54.94 | 57.62 | 59.41 | 61.66 | 38.80 | 24.09 | 16.54 | 10.11 | 5.21 |
| | VAR-GRAD | 55.03 | 60.36 | 62.59 | 63.16 | 64.85 | 41.71 | 37.04 | 33.24 | 24.84 | 9.23 |
| | VAR-IG | 55.21 | 61.22 | 63.04 | 64.29 | 64.31 | 40.21 | 36.85 | 34.09 | 27.71 | 16.43 |
| | VAR-GB | 47.76 | 53.27 | 56.53 | 58.68 | 61.69 | 38.63 | 24.12 | 16.29 | 10.16 | 5.20 |
| **Food_101** | Random | 68.13 | 73.15 | 76.00 | 78.21 | 80.61 | 80.66 | 78.30 | 75.80 | 72.98 | 68.37 |
| | Sobel | 69.08 | 76.70 | 78.16 | 79.30 | 80.90 | 81.17 | 79.69 | 78.91 | 77.06 | 69.58 |
| | GRAD | 78.82 | 82.89 | 83.43 | 83.68 | 83.88 | 83.79 | 83.50 | 83.09 | 82.48 | 78.36 |
| | IG | 82.35 | 83.80 | 83.90 | 83.99 | 84.07 | 84.01 | 83.95 | 83.78 | 83.52 | 80.87 |
| | GP | 77.31 | 79.00 | 78.33 | 79.86 | 81.16 | 80.06 | 79.12 | 77.25 | 78.43 | 75.69 |
| | SG-GRAD | 83.30 | 83.87 | 84.01 | 84.05 | 83.96 | 83.97 | 84.00 | 83.97 | 83.83 | 83.14 |
| | SG-IG | 83.27 | 83.91 | 84.06 | 84.05 | 83.96 | 83.98 | 84.04 | 84.05 | 83.90 | 82.90 |
| | SG-GB | 71.44 | 75.96 | 77.26 | 78.65 | 80.12 | 78.35 | 76.39 | 75.44 | 74.50 | 69.19 |
| | SG-SQ-GRAD | 73.05 | 79.20 | 80.18 | 80.80 | 82.13 | 79.29 | 75.83 | 64.83 | 38.88 | 8.34 |
| | SG-SQ-IG | 72.93 | 78.36 | 79.33 | 80.02 | 81.30 | 79.73 | 76.73 | 70.98 | 59.55 | 27.81 |
| | SG-SQ-GB | 68.10 | 73.69 | 76.02 | 78.51 | 81.22 | 77.68 | 72.81 | 66.24 | 55.73 | 24.95 |
| | VAR-GRAD | 74.24 | 78.86 | 79.97 | 80.61 | 82.10 | 79.55 | 75.67 | 67.40 | 52.05 | 15.69 |
| | VAR-IG | 73.65 | 78.28 | 79.31 | 79.99 | 81.23 | 79.87 | 76.60 | 70.85 | 59.57 | 25.15 |
| | VAR-GB | 67.08 | 73.00 | 76.01 | 78.54 | 81.44 | 77.76 | 72.56 | 66.36 | 54.18 | 23.88 |
| **ImageNet** | Random | 63.60 | 66.98 | 69.18 | 71.03 | 72.69 | 72.65 | 71.02 | 69.13 | 67.06 | 63.53 |
| | Sobel | 65.79 | 70.40 | 71.40 | 71.60 | 72.65 | 72.89 | 71.94 | 71.61 | 70.56 | 65.94 |
| | GRAD | 67.63 | 71.45 | 72.02 | 72.85 | 73.46 | 72.94 | 72.22 | 70.97 | 70.72 | 66.75 |
| | IG | 70.38 | 72.51 | 72.66 | 72.88 | 73.32 | 73.17 | 72.72 | 72.03 | 71.68 | 68.20 |
| | GP | 71.03 | 72.45 | 72.28 | 72.69 | 71.56 | 72.29 | 71.91 | 71.18 | 71.48 | 70.38 |
| | SG-GRAD | 70.47 | 71.94 | 72.14 | 72.35 | 72.44 | 72.08 | 71.94 | 71.77 | 71.51 | 70.10 |
| | SG-IG | 70.98 | 72.30 | 72.49 | 72.60 | 72.67 | 72.49 | 72.39 | 72.26 | 72.02 | 69.77 |
| | SG-GB | 66.97 | 70.68 | 71.52 | 71.86 | 72.57 | 71.28 | 70.45 | 69.98 | 69.02 | 64.93 |
| | SG-SQ-GRAD | 63.25 | 69.79 | 72.20 | 73.18 | 73.96 | 69.35 | 60.28 | 41.55 | 29.45 | 11.09 |
| | SG-SQ-IG | 67.55 | 68.96 | 72.24 | 73.09 | 73.80 | 70.76 | 65.71 | 58.34 | 43.71 | 29.41 |
| | SG-SQ-GB | 62.42 | 68.96 | 71.17 | 72.72 | 73.77 | 69.74 | 60.56 | 52.21 | 34.98 | 15.53 |
| | VAR-GRAD | 53.38 | 69.86 | 72.15 | 73.22 | 73.92 | 69.24 | 57.48 | 39.23 | 30.13 | 10.41 |
| | VAR-IG | 67.17 | 71.07 | 71.48 | 72.93 | 73.87 | 70.87 | 65.56 | 57.49 | 45.80 | 25.25 |
| | VAR-GB | 62.09 | 68.51 | 71.09 | 72.59 | 73.85 | 69.67 | 60.94 | 47.39 | 35.68 | 14.93 |

Table 2: Average test-set accuracy across 5 independent runs for all estimators and datasets considered. ROAR requires removing a fraction of pixels estimated to be most important. KAR differs in that the pixels estimated to be most important are kept rather than removed. The fraction removed/kept is indicated by the threshold. The estimators we report results for are base estimators (Integrated Gradients (IG), Guided Backprop (GB), Gradient Heatmap (GRAD) and derivative ensemble approaches - SmoothGrad, (SG-GRAD, SG-IG, SG-GB), SmoothGrad-Squared (SG-SQ-GRAD, SG-SQ-IG, SG-SQ-GB) and VarGrad (VAR-GRAD, VAR-IG, VAR-GB).