[Reviews · NeurIPS 2019]

Reviewer 1



Summary --- This paper proposes to evaluate saliency/importance visual explanations by removing "important" pixels and measuring whether a re-trained classifier can still classify such images correctly. Many explanations fail to remove such class-relevant information, but some ensembling techniques succeed by completely removing objects. Those are said to be better explanations. (motivation) The goal of an importance estimator is unclear (also called saliency visualizations, visual explanations, heat maps, etc.). This paper takes the view that important information is that information which a classifier can use to predict the correct label. As a result, we can measure whether an importance estimate is good by measuring how much performance drops when the important pixels are removed from all images in both train and val sets. (approach) For each image in a dataset, estimate the importance of all pixels and remove the most important X% of pixels. Train a new model and compare its performance on the augmented val set to the original model's performance on the original val set. The more performance drops, the better the importance estimator is. Three importance estimators are considered: Integrated Gradients, Guided Backprop, and plain Gradients. Furthermore, ensembled versions of each estimator are also considered using 3 variants of SmoothGrad. (experiments) 1. A synthetic dataset is created where we know which 4 of 16 input dimensions are important. Different importance estimators are compared with and without the proposed ROAR. Only ROAR ranks the importance estimators correctly. 2. All estimators except 2 ensemble variants perform worse than a baseline that assigns random importance. Point 2 holds taking multiple random initializations and image classification datasets into account. (conclusion) Existing importance estimators do a bad job according to the newly proposed ROAR metric, SmoothGrad ensembling can help. Stengths --- In addition to the contributions noted above I'd like to point out that the experiments in this paper seem to take a lot of GPU hours, dataset storage space, and effort to put together. Good job! These seem like a solid set of experiments. Weaknesses --- I think human studies in interpretability research are mis-represented at L59. * These approaches don't just ask people whether they think an approach is trustworthy. They also ask humans to do things with explanations and that seems to have a better connection to whether or not an explanation really explains model behavior. This follows the version of interpretability from [1]. This paper laments a lack of theoretical foundation to interpretability approaches (e.g., at L241,L275-277) and it acknowledges at multiple points that we don't know what ground truth for feature importance estimates should look like. Doesn't a person have to interpret an explanation at some point a model for it to be called interpretable? It seems like human studies may offer a way to philosophically ground interpretability, but this part of the paper mis-represents that research direction in contrast with its treatment of the rest of the related work. Minor evaluation problems: * Given that there are already multiple samples for all these experiments, what is the variance? How significant are the differences between rankings? I only see this as a minor problem because the differences on the right of figure 4 are quite large and those are what matter most. * I understand why more baseline estimators weren't included: it's expensive. It would be interesting to incorporate lower frequency visualizations like Grad-CAM. These can sometimes give significantly different performance (e.g., as in [3]). I expect it may have significant impact here because a more coarse explanation (e.g., 14x14 heatmap) may help avoid noise that comes from the non-smooth, high frequency, per-pixel importance of the explanations investigated. This seems further confirmed by the visualizations in figure 1 which remove whole objects as pointed out at L264. The smoothness of coarse visualization method seems like it should do something similar, so it would further confirm the hypothesis about whole objects implied at L264. * It would be nice to summarize ROAR into one number. It would probably have much more impact that way. One way to do so would be to look at the area under the test accuracy curves of figure 4. Doing so would obscure richer insights that ROAR would provide, but this is a tradeoff made by any aggregate statistic. Presentation: * L106: This seems to carelessly resolve a debate that the paper was previously careful to leave open (L29). Why can't it be that the distribution has changed? Do any experiments disentangle changes in distribution from removal of information? Things I didn't understand: * L29: I didn't get this till later in the paper. I think I do now, but my understanding might change again after the rebuttal. More detail here would be useful. * L85: Wouldn't L1 regularization be applied to the weights? Is that feature selection? What steps were actually taken in the experiments used in this paper? Did the ResNet50 used have L1 regularization? * L122: What makes this a bit unclear is that I don't know what is and what is not a random variable. Normally I would expect some of these (epsilon, eta) to be constants. Suggestions --- * It would be nice to know a bit more about how ROAR is implemented. Were the new datasets dynamically generated? Were they pre-processed and stored? * Say you start re-training from the same point. Train two identical networks with different random seeds. How similar are the importance estimates from these networks (e.g. using rank correlation similarity)? How similar are the sets of the final 10% of important pixels identified by ROAR across different random seeds? If they're not similar then whatever importance estimator isn't even consistent with itself in some sense. This could be thought of as an additional sanity check and it might help understand why the baseline estimators considered don't do well. [1]: Doshi-Velez, F., & Kim, B. (2017). A Roadmap for a Rigorous Science of Interpretability. ArXiv, abs/1702.08608. Final Evaluation --- Quality: The experiments were thorough and appropriately supported the conclusions. The paper really only evaluate importance estimators using ROAR. It doesn't really evaluate ROAR itself. I think this is appropriate given the strong motivation the paper has and the lack of concensus about what methods like ROAR should be doing. Clarity: The paper could be clearer in multiple places, but it ultimately gets the point across. Originality: The idea is similar to [30] as cited. ROAR uses a similar principle with re-training and this makes it new enough. Significance: This evaluation could become popular, inspire future metrics, and inspire better importance estimators. Overall, this makes a solid contribution. Post-rebuttal Update --- After reading the author feedback, reading the other reviews, and participating in a somewhat in-depth discussion I think we reached some agreement, though not everyone agreed about everything. In particular, I agree with R4's two recommendations for the final version. These changes would address burning questions about ROAR. I still think the existing contribution is a pretty good contribution to NeurIPS (7 is a good rating), though I'm not quite as enthusiastic as before. I disagree somewhat with R4's stated main concern, that ROAR does not distinguish enough between saliency methods. While it would be nice to have more analysis about the differences between these methods, ROAR is only one way to analyze these explanations and one analysis needn't be responsible for identifying differences between all the approaches it analyzes.

Reviewer 2



The paper's idea of the benchmarking salience methods on retrained models is simple, original, and clever. It addresses one of the basic problem with evaluating salience of pixel maps, which is that the pixels are not independent of one another. ROAR elegantly sidesteps this problem by proposing simply evaluating a retrained model on a dataset in which estimated salient pixels have been grayed out: if redundant information remains, the retrained model will find it and use it. However, the method raises a few key questions, that if answered, could make this a stronger paper. 1. What is actually being measured by ROAR? While the proposed test is an interesting setup, it seems different from the original question tackled by salience maps, which is "how is the model that I have making its decisions?" Since ROAR retrains (many) new models, it is not testing faithfulness of a salience method the original model that I began with. The paper would be improved if it stated more clearly what ROAR is actually aiming to measure. 2. Specifically, will ROAR penalize a salience method for identifying a biased sensitivity (e.g., from a model that attends to only corner pixels in decoy mnist [AS Ross 2017 "Right for the right reasons"]). That is, would ROAR incorrectly score such a faithful salience model that highlights the model's erroneous attenion on decoy inputs as less accurate than one that also highlights the redundant main input pixels? Another question: 3. I am confused by the warning on lines 84-87 of page 3. My understanding of the method as described is that the most salient pixels are replaced by a gray average over the entire training set, and then after this data erasure, the model is retrained several times over this modified data set. Does something different have to be done when a model with L1 regularization is used?

Reviewer 3



UPDATED REVIEW AFTER REBUTTAL AND DISCUSSION: After discussing with the other reviewers, I'm increasing my rating. However, my concerns still stand (and are shared by the other reviewers to different degrees); the paper can be greatly improved if these are addressed in the final version. My main concern is as follows: there's little in the paper that helps the reader analyze the results of their experiments and the meaning of their method: In lines 260-267; pg 8: the authors report that all saliency methods roughly perform the same, with the exception of SG-SQ-GRAD and VAR-GRAD. They explain the similarity between SG-SQ-GRAD and VAR-GRAD; however, they fail to explain substantively how these methods differ from the rest (besides mentioning that they capture whole objects better) or why the other methods are so indistinguishable from each other. In particular, ROAR may be vulnerable / bias to the "style" of a saliency method (i.e., coarseness vs. finegrained-ness, as represented by CAM/Grad-CAM vs. gradient). R1: "1. Experiments that evaluate a coarse importance estimator" R2: "1. A clearer statement of what ROAR is trying to measure (and ideally experimental evidence to back it up) and how it might differ from the problem traditionally tackled by salience maps." I'd recommend the authors do the following for the final version: 1. discuss and compare against the existing post-hoc deletion game, i.e., like the one in RISE (this is fairly easy to do, as no re-training is required) 2. discuss (at the very least) and analyse whether / to what extent ROAR is vulnerable or bias to the "style" of the saliency method (i.e., try a coarse method like CAM/Grad-CAM, as initially suggested by R1 or add a "patch extraction" step that fixes feature coarseness across all methods) -- this can be a single experiment (not an exhaustive addition of another saliency method to the existing experiments) Lastly, I did not ask for evaluation on more datasets and methods (as the authors claimed in their rebuttal). My only related mention of this is in the context of my second suggestion as a way to iterate more quickly in order to improve the technical aspects of the method (i.e., use a smaller dataset / easier task such as fine-tuning; and that using a quicker iteration pipeline would allow for evaluation of more methods). --- This paper introduces ROAR (RemOve And Retrain), a novel metric for evaluating saliency methods. ROAR involves removing a percentage of features identified as "salient" and retraining a model with these features removed. The authors provide strong motivation for ROAR in section 3 and Figure 2; principally, this is the typical "deletion metric" used which involves evaluating models on images with removed features causes models to evaluate on out-of-training-domain distribution (i.e., images with removed features). The main drawback of this work is that it presents negative results, in that it is generally unable to discriminate the quality of different saliency methods (with the exception of SmoothGrad-Squared and VarGrad). This work would benefit from more thought and improvement on the metric itself to diagnose why it's currently not discriminative. The significance of this work is severely hampered by the negative results; I would only change my score if the method is remedied to demonstrate empirical value. Generally, the paper is clearly written.

[Author Response · NeurIPS 2019]

We thank the reviewers for their time and thoughtful comments.

**re: L1 regularization. (R1 and R2)** Our implementation did not use L1 reg. We added this note to describe a variation of ROAR on models with explicit feature selection, e.g. L1 regularization in a linear model. Here we know which features are not used, hence we can prevent them from being used when retraining. This makes ROAR more reliable.

**Reviewer 1 (R1)  re: portrayal of human studies:** R1 correctly points out our portrayal of human studies requires more nuance. We would be glad to correct this and will update the manuscript accordingly.
**re: variance of the results:** The variance is very low across all datasets and estimators. The maximum variance observed for ImageNet was a variance of 1.32% using SG-SQ-GRAD at 70% of inputs removed. On Birdsnap the highest variance was 0.12% using VAR-GRAD at 90% removed. For food101 it was 1.52% using SG-SQ-GRAD at 70% removed. As the reviewer assumed correctly, the gap between estimators is far larger than the variance.
**re: single ROAR metric using AUC:** This is something we will consider when benchmarking additional methods in the future. But as the reviewer points out, sometimes the curve itself provides additional information.
**re: ROAR dataset generation:** The saliency maps for the datasets were pre-computed and stored on disk. These were then combined in the pre-processing pipeline to mask out the required part of the image.

**Reviewer 2 (R2)  re: What ROAR measures:** At line 101, we discuss the nuances of using a set of ROAR models to evaluate the accuracy of an explanation produced on a different original model. We will update the manuscript to discus this earlier, so it is clearer to the reader what ROAR measures.
**re: Decoy MNIST:** In the Decoy MNIST training set, the corners of the image are modified to be predictive of the label. On the test set, these modifications are random. If the model uses these corners the test set accuracy will be low. Assuming the patches are the most important and the interpretability method detects this, the following happens using ROAR. As the corners are masked, test set performance increases. After this the accuracy would start degrading. A random baseline would be expected to exhibit a steady decrease. We believe that with a comparison to the random baseline the odd behavior can be detected (and explained).

**Reviewer 4 (R4)**  R4 argues that "*The main drawback of this work is that it presents negative results*", however this manuscript results in a strong **positive recommendation** for the use of SmoothGrad-Squared and VarGrad. We find that these ensemble estimators *far* outperform all other methods; for example on ImageNet, there is a remarkable gap of 56.34% between the best performing ensemble method (VarGrad) and the best performing base methods (GRAD). The result is consistent across all 3 large scale datasets we evaluate.

The **significance** of the work is also supported by the other reviewers. **R1:** "*This evaluation could become popular, inspire future metrics, and inspire better importance estimators.*". **R2:** "*Surprisingly, most methods underperform random feature ablation, and also surprisingly smoothgrad-squared and a similar method far outperform the other methods. This finding raises interesting questions about both the failure of many traditional methods as well as how smoothgrad-squared works so well. This is a very interesting result that will lead to follow-on work, and it is a second significant contribution.*".

re: "*whether or not curves [...] with and without retraining generally match*". We note that this experiment is performed on toy data in **Fig. 2** and on Imagenet in **Fig. 3**, and the results strongly support our stated methodology of re-training. Without retraining, the 'removal' of pixels by replacing with a constant introduces new image statistics that were not seen by the model during training.

R4 suggest we evaluate on **more datasets and methods**. We note that we already present consistent results on both toy data and several large scale, natural image datasets such as ImageNet, BirdSnap and Food 101. In addition to two baselines (sobel edge detector and random), we compare 12 methods in the main paper (and 4 in suppl.). In total, we generate 540 large-scale modified image datasets in order to consider all experiment variants (180 new test/train for each original dataset). For each of these datasets, we independently train 5 ResNet-50 models from random initialization.

**re: Fong and Veldadi**, who propose a minimum perturbation area which preserves the model prediction. This minimum deletion area is identified by perturbing and evaluating the model output without retraining. The authors openly acknowledge the shortcomings of not re-training is the introduction of artefacts that could change the image to be out of distribution. This is *precisely* the reason we do insist upon retraining on the modified inputs, such that the model can learn that the constant we use to replace the pixels removed are uninformative.

**re: considering dropping patches vs. individual pixels**. We note that Liu et al. had a different stated motivation of image in-painting. However, we agree that evaluating rankings according to different units of importance would be valuable. Due to computational constraints, we leave this as the subject for future work.

[Meta-Review · NeurIPS 2019]

Dear authors, congrats on the acceptance-- this paper was discussed extensively and controversially. In balance, we decided that the straightforward metric and empirical results will be interesting for Neurips, although at the same time it is clear that there is work to be done: The reviewers provided multiple comments and a lot of feedback, and it will be of critical importance that you revise the manuscript accordingly (and as you promised in your rebuttal)- in particular, this would require significant changes in the exposition, and also a more clear description of the novelty of the study in light of previous work (in addition to the suggestions by the reviewers, it strikes me that the classic literature on recursive feature elimination could really be mentioned at least).